# Contribution of *Trp63*[CreERT2]-labeled cells to alveolar regeneration is independent of tuft cells

Huachao Huang[1,2], Yinshan Fang[1,2], Ming Jiang[3], Yihan Zhang[4], Jana Biermann[5,6], Johannes C Melms[5,6], Jennifer A Danielsson[7], Ying Yang[8], Li Qiang[9], Jia Liu[10], Yiwu Zhou[11], Manli Wang[10], Zhihong Hu[10], Timothy C Wang[2], Anjali Saqi[9], Jie Sun[12], Ichiro Matsumoto[13], Wellington V Cardoso[1,14], Charles W Emala[7], Jian Zhu[15], Benjamin Izar[5,6], Hongmei Mou[4]*, Jianwen Que[1,2]*

[1]Columbia Center for Human Development, Department of Medicine, Columbia University Medical Center, New York, United States; [2]Division of Digestive and Liver Diseases, Department of Medicine, Columbia University Medical Center, New York, United States; [3]Institute of Genetics, the Children's Hospital, Zhejiang University School of Medicine, Hangzhou, China; [4]Mucosal Immunology and Biology Research Center, Massachusetts General Hospital, Harvard Medical School, Boston, United States; [5]Division of Hematology/Oncology, Department of Medicine, Columbia University Irving Medical Center, New York, United States; [6]Columbia Center for Translational Immunology, Columbia University Irving Medical Center, New York, United States; [7]Department of Anesthesiology, Columbia University Irving Medical Center, New York, United States; [8]Program in Epithelial Biology, Stanford University School of Medicine, Stanford, United States; [9]Department of Pathology & Cell Biology, Columbia University Medical Center, New York, United States; [10]State Key Laboratory of Virology, Wuhan Institute of Virology, Center for Biosafety Mega-Science, Chinese Academy of Sciences, Wuhan, China; [11]Department of Forensic Medicine, Tongji Medical College of Huazhong University of Science and Technology, Wuhan, China; [12]Carter Immunology Center, the University of Virginia, Charlottesville, United States; [13]Monell Chemical Senses Center, Philadelphia, United States; [14]Division of Pulmonary, Allergy and Critical Care Medicine, Department of Medicine, Columbia University Medical Center, New York, United States; [15]Department of Pathology, Ohio State University College of Medicine, Columbus, United States

*For correspondence:
HMOU@mgh.harvard.edu (HM);
jq2240@columbia.edu (JQ)

**Competing interest:** The authors declare that no competing interests exist.

**Abstract** Viral infection often causes severe damage to the lungs, leading to the appearance of ectopic basal cells (EBCs) and tuft cells in the lung parenchyma. Thus far, the roles of these ectopic epithelial cells in alveolar regeneration remain controversial. Here, we confirm that the ectopic tuft cells are originated from EBCs in mouse models and COVID-19 lungs. The differentiation of tuft cells from EBCs is promoted by Wnt inhibition while suppressed by Notch inhibition. Although progenitor functions have been suggested in other organs, pulmonary tuft cells don't proliferate or give rise to other cell lineages. Consistent with previous reports, *Trp63*[CreERT2] and *KRT5-CreERT2*-labeled ectopic EBCs do not exhibit alveolar regeneration potential. Intriguingly, when tamoxifen was administrated post-viral infection, *Trp63*[CreERT2] but not *KRT5-CreERT2* labels islands of alveolar epithelial cells that are negative for EBC biomarkers. Furthermore, germline deletion of *Trpm5* significantly increases the contribution of *Trp63*[CreERT2]-labeled cells to the alveolar epithelium. Although Trpm5 is known to regulate tuft cell development, complete ablation of tuft cell production fails to improve alveolar

regeneration in *Pou2f3*<sup>-/-</sup> mice, implying that Trpm5 promotes alveolar epithelial regeneration through a mechanism independent of tuft cells.

## Editor's evaluation

In this manuscript, the authors describe the ectopic tuft cells that were derived from lineage-tagged Krt5+ Trp63+ cells post influenza virus infection. These tuft cells do not appear to proliferate or give rise to other lineages in the lung parenchyma. They then claim that Wnt inhibitors increase the number of tuft cells while inhibiting Notch signalling decreases the number of tuft cells within Krt5+ pods after infection in vitro and in vivo. The authors further show that genetic deletion of Trpm5 results in an increase in AT2 and AT1 cells in p63CreER lineage-tagged cells compared to control. Lastly, they demonstrate that depletion of tuft cells caused by genetic deletion of Pou2f3 has no effect on the derivation of AT1 and AT2 cells from the p63+ Krt5- progenitor population, implying that tuft cells play no functional role in this process. This study provides new insights into the role of tuft cells and p63+Krt5- progenitor cells in lung repair.

## Introduction

Postmortem examination of H1N1 influenza-infected lungs revealed extensive tissue remodeling accompanied by the presence of ectopic cytokeratin-5-positive (KRT5$^+$) basal cells in the lung parenchyma (*Xi et al., 2017*). These ectopic basal cells (EBCs) are also present in the mouse parenchyma following infection with modified H1N1 influenza PR8 virus or treatment with a high dose of bleomycin (*Kanegai et al., 2016*; *Vaughan et al., 2015*; *Xi et al., 2017*; *Yuan et al., 2019*; *Zacharias et al., 2018*; *Zuo et al., 2015*). Initial studies suggest that these EBCs contribute to alveolar regeneration (*Zuo et al., 2015*). However, studies from other groups suggest that EBCs do not meaningfully become alveolar epithelial cells, but rather provide structural supports to prevent the lung from collapsing (*Basil et al., 2020*; *Kanegai et al., 2016*; *Vaughan et al., 2015*).

Tuft cells are a minor cell population critical for chemosensory and relaying immune signals in multiple organs including the intestine and trachea (*Howitt et al., 2016*; *Montoro et al., 2018*; *von Moltke et al., 2016*). In the intestine tuft cells serve as immune sentinels during parasitic infection (*Gerbe et al., 2016*; *Howitt et al., 2016*; *McGinty et al., 2020*; *von Moltke et al., 2016*). Responding to helminth infection, tuft cells release the alarmin interleukin (IL)–25 which activates type 2 innate lymphoid cells (ILC2s) and their secretion of IL-13, initiating type 2 immune response to eliminate parasitic infection (*Gerbe et al., 2016*; *Howitt et al., 2016*; *von Moltke et al., 2016*). Tuft cells were initially identified in the rat trachea over six decades ago (*Rhodin and Dalhamn, 1956*). However, we just have begun to appreciate their functions in the respiratory system (*Bankova et al., 2018*; *Ualiyeva et al., 2020*). Recent single-cell RNA sequencing confirmed the presence of tuft cells in the mouse trachea where they express several canonical genes, such as *Alox5ap*, *Pou2f3*, and *Gfi1b* (*Montoro et al., 2018*). Following repeated allergen challenges, tuft cells expand and amplify immune reactions (*Bankova et al., 2018*). Intriguingly, tuft cells were found ectopically present in the lung parenchyma, co-localized with EBCs following PR8 viral infection (*Rane et al., 2019*). Although p63-expressing lineage-negative epithelial stem/progenitor cells (LNEPs) were considered as a source of tuft cells (*Rane et al., 2019*), it is still unclear whether these p63$^+$ cells immediately give rise to tuft cells or through EBCs. More recently, we reported that ectopic tuft cells were also present in the parenchyma of COVID-19 lungs, and ablation of tuft cells dampens macrophage infiltration at the acute phase following viral infection in a mouse model (*Melms et al., 2021*). However, the role of tuft cells in lung regeneration following viral infection remains undetermined.

In this study, we showed that EBCs served as the cell of origin for the ectopic tuft cells present in the parenchyma during viral infection. Upon screening multiple signaling pathways we identified that Notch inhibition blocked tuft cell derivation, while Wnt inhibition significantly enhanced tuft cell differentiation from EBCs. We then used multiple mouse models to demonstrate that *Trp63*$^{CreERT2}$ labeled subpopulations of alveolar type 1 and 2 (AT1/AT2) cells during regeneration regardless of the presence of tuft cells in the parenchyma.

## Results

### Tuft cells expanded in response to challenges with influenza, bleomycin, or naphthalene

Tuft cells are rarely present in the proximal airways of normal adult mice (*Montoro et al., 2018*). However, they were present in the distal airways at birth when tuft cells were first detected (*Figure 1A and B*). Tuft cells continued to be present throughout the airways when examined at postnatal (P)10 and P20 but were no longer detected in the terminal bronchiole at P56 (*Figure 1A and B*). Notably, tuft cells reappeared in the terminal airways approximately 7 days post infection (dpi) with PR8 virus (*Figure 1C*). The number of tuft cells also significantly increased in the large airways at 30 dpi (4.9±1.1 vs 25.7±3.3 per 1000 epithelial cells) (*Figure 1C*). As previously described, tuft cells were present in the EBC area (*Melms et al., 2021*; *Rane et al., 2019*). Further analysis revealed that tuft cells were first detected in the EBC area at approximately 15 dpi and peaked at 21 dpi (*Figure 1—figure supplement 1*). The numbers of tuft cells also expanded in the large airways following naphthalene (13.4±1.1 per 1000 epithelial cells) or bleomycin challenge (8.6±0.8 per 1000 epithelial cells) (*Figure 1D*). In addition, tuft cells were ectopically present within EBCs 28 days following challenge with a high dose of bleomycin (*Figure 1D*). These findings suggest that tuft cell expansion in the airways is a general response to lung injuries, and that severe injuries induce ectopic tuft cells in the parenchyma.

### The ectopic tuft cells in the parenchyma did not proliferate or give rise to other cell lineages

Previous studies suggest that Dclk1[+] cells serve as progenitor cells in the small intestine and pancreas (*May et al., 2008*; *May et al., 2009*; *Westphalen et al., 2016*). We asked whether the ectopic tuft cells in the lung parenchyma also possess progenitor potential. We performed lineage tracing with a knock-in *Pou2f3^{CreERT2}* mouse line (*McGinty et al., 2020*). Tuft cells (Dclk1[+]) in the large airways were specifically labeled with the lineage tracing maker (tdT[+]) upon three Tamoxifen (Tmx) injections into *Pou2f3^{CreERT2};R26^{Ai14}* mice (*Figure 2A*). Notably,~92% tdT[+] cells expressed Dclk1 in the trachea, whereas all tdT[+] cells expressed Dclk1 in the large intrapulmonary airways (*Figure 2A* and *Figure 2—figure supplement 1A*). We then analyzed the mice challenged with PR8 virus. Tmx was continuously given from 14 to 27 dpi and the lungs were analyzed at 60 dpi (*Figure 2B*). The majority (~93%) of the ectopic tuft cells were lineage labeled in the lung parenchyma of *Pou2f3^{CreERT2};R26^{Ai14}* mice (*Figure 2B*). These labeled tuft cells solitarily distributed in the injured lung parenchyma (*Figure 2B*), indicating that they had not undergone expansion. Consistently, none of the tuft cells was positive for the proliferation marker Ki67 (*Figure 2—figure supplement 1B*). To further test whether tuft cells give rise to other cell lineages, immunostaining with various cell type markers was performed, including Scgb1a1 (club cells), FoxJ1 (ciliated cells), and Clca3 (goblet cells). The results showed that tuft cells did not contribute to other cell types following a chasing period up to 60 days (*Figure 2C*). These data suggest that tuft cells in the lung unlikely serve as progenitor cells.

### The ectopic tuft cells in the parenchyma originated from EBCs in PR8-infected mice and COVID-19 lungs

Basal cells serve as the origin of tuft cells in the mouse trachea (*Montoro et al., 2018*). By contrast, p63-expressing lineage-negative epithelial stem/progenitor cells (LNEPs) were considered as the origin of tuft cells in the lung parenchyma following viral infection (*Rane et al., 2019*). While characterizing ectopic tuft cells in the parenchyma, we noticed approximately 5% tuft cells co-expressed Dclk1 and the basal cell marker Krt5 (*Figure 2—figure supplement 1C*), suggesting that these tuft cells are in a transitioning state from EBCs towards tuft cells. To test this hypothesis, *Trp63^{CreERT2};R26^{Ai14}* mice were infected with PR8 virus followed by daily Tmx injection from 14 dpi to 18 dpi as tuft cells were initially detected at around 15 dpi (*Figure 2D*). The majority of tuft cells within EBCs were labeled with tdT (*Figure 2D*). We also used *KRT5-CreERT2;R26^{Ai14}* mice to trace EBCs-derived tuft cells. Consistently, about 80% of tuft cells were labeled with tdT (*Figure 2E*). These findings confirmed that EBCs serve as the cell origin for the ectopic tuft cells. EBCs can be derived from multiple cell sources including a subpopulation of Scgb1a1[+] club cells (*Yang et al., 2018*). To test whether EBCs derived from the club cell subpopulation give rise to tuft cells, we lineage labeled club cells before exposing *Scgb1a1^{CreERT};R26^{Ai14}* mice to PR8 virus. 21.0% ± 2.0% tuft cells that were ectopically present

**Figure 1.** Tuft cells in homeostasis and their expansion in response to severe injuries. (**A**) Tuft cells are present in the distal airways at P0, P10, P20 but not P56. n=4 for each group. (**B**) Quantification of tuft cells from panel (**A**). (**C**) Tuft cells expand in the large airway and are ectopically present in the terminal airways following viral infection. (**D**) Tuft cells expand in the airways following naphthalene and bleomycin challenge and are ectopically present in parenchyma following bleomycin challenge. Scale bars, 50 μm.

*Figure 1 continued on next page*

*Figure 1 continued*

The online version of this article includes the following figure supplement(s) for figure 1:

**Figure supplement 1.** Ectopic tuft cells appear after viral infection.

in the parenchyma were labeled with tdT when examined at 30 dpi (*Figure 2—figure supplement 1D*), suggesting that club cells also generate a portion of tuft cells in the lung parenchyma upon viral infection.

We recently reported expansion of tuft cell in COVID-19 lungs (*Melms et al., 2021*). Upon SARS-CoV-2 infection club and ciliated cells were almost completely ablated (*Fang et al., 2020*), exposing the underlying basal cells in the affected intrapulmonary airways (*Figure 3A*). A significant number of the dispositioned basal cells remained proliferative while lodged in the alveoli, presumably initiating EBCs (*Figure 3B–C*). Tuft cells were occasionally present in the established EBCs in addition to the airways (*Figure 3D*). Similar to the EBC-derived tuft cells in influenza-infected mice, the ectopic tuft cells in the parenchyma of COVID-19 lungs co-expressed KRT5 and POU2F3 (*Figure 3D*), suggesting a similar differentiation scheme. By contrast, tuft cells in the airways did not express KRT5 (*Figure 3E*).

## Inhibition of Wnt and Notch signaling has opposite effects on the differentiation of EBCs into tuft cells

We next sought to identify the signaling pathways that can influence tuft cell differentiation from EBCs. Lineage-labeled EBCs were isolated and purified from the lungs of *KRT5-CreERT2;R26$^{Ai14}$* mice following viral infection and were expanded using the protocol for culturing basal cells (*Mou et al., 2016*; *Figure 4—figure supplement 1A*). The expanded EBCs maintained the lineage tag (tdT$^+$) and expressed p63 and Krt5 (*Figure 4—figure supplement 1B*). In addition, they also expressed the respiratory protein markers Nkx2.1 and Foxa2 (*Figure 4—figure supplement 1B*). We then used the expanded EBCs to assess the impact of major signaling pathways which have been implicated in the determination of cell fate during lung development. The tested pathways include Tgfß/Bmp, Yap, Shh, Wnt, Notch, IL-6, and IL-4/IL-13 (*Ahdieh et al., 2001*; *Barkauskas et al., 2013*; *Ikonomou et al., 2020*; *Lee et al., 2014a*; *Li et al., 2017*; *Nabhan et al., 2018*; *Pardo-Saganta et al., 2015*; *Tadokoro et al., 2014*; *Vaughan et al., 2015*; *Yuan et al., 2019*; *Zacharias et al., 2018*). We observed that treatment with IL-4 or IL-13 promoted tuft cell derivation from EBCs in air-liquid interface (ALI) culture (data not shown), in agreement with the previous reports of IL-4Rα-dependent tuft cell expansion from the intestinal crypts (*Gerbe et al., 2016*; *von Moltke et al., 2016*). Notably, treatment of the WNT signaling activator CHIR9902 blocked the derivation of tuft cells by 35.3% ± 3.1%. Conversely, treatment with the Wnt inhibitor IWR-1 promoted tuft cell differentiation by 32.4% ± 6.4% (*Figure 4—figure supplement 1C*). Inhibition of WNT signaling with the Porcupine inhibitor Wnt-C59 also led to increased tuft cell differentiation of airway basal cells isolated from *Dclk1-GFP* mice (*Figure 4A*). Consistently, daily injection of Wnt-C59 induced abundant tuft cells in the lung parenchyma following viral infection (*Figure 4B*). By contrast, Notch inhibition with the γ-secretase inhibitor Dibenzazepine (DBZ) reduced tuft cell differentiation from EBCs in ALI culture (*Figure 4C*). Daily DBZ injection also decreased the number of tuft cells in the parenchyma by 63.6% ± 1.5% following viral infection (*Figure 4D*). To further confirm the role of Notch signaling in tuft cell derivation, *KRT5-CreERT2;Rbpjk$^{f/f}$;R26$^{Ai14}$* mice were infected with PR8 virus and injected with Tmx daily from 14 dpi to 18 dpi to specifically delete the Notch effector *Rbpjk* in EBCs. Consistent with the in vitro finding, tuft cells were barely present in the lung parenchyma of *KRT5-CreERT2;Rbpjk$^{f/f}$;R26$^{Ai14}$* mice (*Figure 4—figure supplement 1D*). Together these findings suggest that inhibition of the WNT signaling pathway promotes while Notch signaling inhibition blocks the differentiation of EBCs into tuft cells.

## Enhanced generation of *Trp63$^{CreERT2}$* lineage labeled alveolar epithelium in *Trpm5$^{-/-}$* but not *Pou2f3$^{-/-}$* mutants

Given tuft cell ablation reduces macrophage infiltration in PR8-infected mouse lungs (*Melms et al., 2021*), we asked whether loss of tuft cells improves alveolar regeneration. The first mouse model we examined was *Trpm5* null mutant which demonstrates ~80% reduction of tuft cells in the intestine (*Howitt et al., 2016*). The number of tuft cells was reduced but not significantly following viral infection in the lung parenchyma of *Trp63$^{CreERT2}$;Trpm5$^{-/-}$;R26$^{Ai14}$* mice as compared to control mice



**Figure 2.** Tuft cells in the parenchyma are derived from ectopic basal cells (EBCs) following H1N1 PR8 viral infection. (**A**) *Pou2f3^{CreERT2};R26^{Ai14}* mouse line specifically labels Dclk1+ tuft cells in the large airways at homeostasis. (**B**) Lineage tracing of Pou2f3+ cells in the parenchyma after PR8 virus infection. Arrow indicates a tdT+Dclk1- cell. (**C**) Lineage labeled tuft cells do not contribute to other cell lineages, including Scgb1a1+ club cells, FoxJ1+ ciliated cells and Clca3+ goblet cells. (**D**) Lineage tracing confirms that tuft cells are derived from EBCs in the lung parenchyma of *Trp63^{CreERT2};R26^{Ai14}* mice

*Figure 2 continued on next page*

*Figure 2 continued*

following PR8 infection. (**E**) Lineage tracing shows about 80% of tuft cells are derived from EBCs in the lung parenchyma of *KRT5-CreERT2;R26^{Ai14}* mice following PR8 infection. Data represent mean ± s.e.m. Scale bars, 20 µm (**A**) and 50 µm (B to G).

The online version of this article includes the following figure supplement(s) for figure 2:

**Figure supplement 1.** Tuft cells in the lung parenchyma are derived from ectopic basal cells (EBCs).

---

(*Trp63^{CreERT2};Trpm5^{+/-};R26^{Ai14}*) (*P*>0.05, ***Figure 5—figure supplement 1A***). We did not observe any apparent reduction in the size of Krt5⁺ EBC clones in the lung parenchyma (***Figure 5—figure supplement 1B***). Consistently, *KRT5-CreERT2* lineage labeled cells did not contribute to AT1 or AT2 cells no matter Tmx was administered before or after viral infection (data not shown), which is in line with previous studies (***Kanegai et al., 2016***; ***Vaughan et al., 2015***). Surprisingly, when Tmx was injected after viral infection, we observed 7.9% ± 1.4% *Trp63^{CreERT2}* lineage-labeled cells expressed the AT2 cell marker SftpC but not p63 or Krt5 in the areas that surrounded the injured foci, and the number was increased to 19.4%±4.2% in *Trp63^{CreERT2};Trpm5^{-/-};R26^{Ai14}* mice (***Figure 5A*** and ***Figure 5— figure supplement 1C***). Lineagelabeled AT2 cells were further confirmed by Abca3, another mature AT2 cell marker (***Figure 5—figure supplement 1D***). In addition, lineage-labeled cells also included AT1 epithelium (Hopx⁺, T1α⁺) which was increased in the parenchyma of *Trp63^{CreERT2};Trpm5^{-/-};R26^{Ai14}* mutants (6.5% ± 0.9% vs 2.5% ± 0.8% at 60 dpi) (***Figure 5B*** and ***Figure 5—figure supplement 1E***). To assess whether *Trpm5^{-/-}* mice exhibited improved pulmonary mechanics after viral infection, we assessed airway resistance when challenged with increasing concentrations of methacholine. Total respiratory system resistance (Rrs) and central airway resistance (Rn) were significantly attenuated in *Trpm5^{-/-}* mice (***Figure 5C***).

We next examined the contribution of *Trp63^{CreERT2}* lineage-labeled cells to lung regeneration in *Pou2f3^{-/-}* mice which have no tuft cells (***Matsumoto et al., 2011***). We subjected *Trp63^{CreERT2};Pou2f3^{-/-};R26^{Ai14}* mutants and controls (*Trp63^{CreERT2};Pou2f3^{+/-};R26^{Ai14}*) to viral infection followed by continuous Tmx injection from 14 to 18 dpi. In the controls prominent bronchiolization occurred in the parenchyma with the extensive presence of Krt5⁺ EBCs (***Figure 6A***), and approximately 8% of the lineage-labeled cells co-expressed Sftpc but not Krt5 (***Figure 6B***). We did not detect improved contribution of lineage-labeled alveolar epithelium in the mutants as compared to the controls (p>0.05) (***Figure 6B***), suggesting that loss of the ectopic tuft cells has no impact on alveolar regeneration initiated by *Trp63^{CreERT2}* labeled epithelium.

## Discussion

Tuft cells are observed in the large airways at homeostasis in adults. Here, we showed that tuft cells are present in both large and terminal airways during early postnatal development. In response to severe injuries, tuft cells expanded in the airways and ectopically presented in the parenchyma of severely injured lungs. These tuft cells did not generate other cell lineages. Lineage tracing confirmed that they were derived from EBCs, which was promoted by Wnt inhibition. By contrast, pharmacological or genetic inhibition of Notch signaling blocked EBC differentiation into tuft cells. We confirmed that EBCs (Krt5⁺, p63⁺) do not contribute to alveolar regeneration. Instead, we found a *Trp63^{CreERT2}* labeled population generated alveolar cells independent of the presence of tuft cells.

Tuft cells were detected throughout the airways including the terminal airways at around the neonatal stage, and they were no longer present in the terminal airways when examined at P56. Upon viral infection, tuft cells expanded in large airways and re-appeared in terminal airways, suggesting re-activation of developmental signaling during injury-repair. Our pilot screen of signaling inhibitors/activators demonstrated that Wnt blockage resulted in significantly increased differentiation of EBCs towards tuft cells. In consistence, treatment of the Wnt inhibitor Wnt-C59 also led to significant expansion of tuft cells in the airways and parenchyma. Along this line, influenza infection has been shown to cause downregulation of Wnt signaling in mouse lungs (***Hancock et al., 2018***). By contrast, we found that both pharmacological and genetic inhibition of Notch signaling suppressed tuft cell expansion. Additionally, IL-4 and IL-13 treatments increased the derivation of tuft cells from EBCs, which is in contrast to the findings from the co-submitted manuscript where deletion of *Il4* had no impact on tuft cell differentiation. This could be due to the extra IL-4/IL-13 we supplied to the cell culture, which may not be present in an in vivo setting during viral infection. Re-expression of the transcription factor

**Figure 3.** EBCs likely give rise to tuft cells in the parenchyma of COVID-19 lungs. (**A**) SARS-CoV-2 infection causes the loss of club and ciliated cells (arrows in a), exposing the underlying basal cells (B) in the small airways. Note the detached basal cells (**C**). (**B**) EBCs proliferate in the parenchyma of COVID-19 lungs. H&E staining shows the presence of EBC clusters in COVID-19 lungs. (**C**) Representative clusters of EBCs are present in COVID-19 lungs. (**D**) Tuft cells within EBCs express both KRT5 and the tuft cell marker POU2F3. (**E**) Solitary tuft cells without KRT5 expression are present in the airways of COVID-19 lung. Abbreviation: bv, blood vessel. Scale bars, 100 µm (**A, B and C**) and 50 µm (**D and E**).

Sox9 has been observed during the repair of the airway epithelium following naphthalene challenge (*Jiang et al., 2021*). Interestingly, Sox9 is expressed by tuft cells as shown by the accompanied manuscript. It will be interesting in future experiments to determine whether Sox9 is required to promote the generation of tuft cells.

**Figure 4.** Wnt inhibition promotes EBC differentiation into tuft cells while Notch inhibition has an opposite effect. (**A**) Treatment with the WNT signaling inhibitor Wnt-C59 promotes tuft cell differentiation of *Dclk1-GFP* basal cells in ALI culture. (**B**) Wnt-C59 treatment increases the number of tuft cells in the lung parenchyma following viral infection. (**C**) Treatment with the Notch signaling inhibitor DBZ completely blocks tuft cell differentiation of

*Figure 4 continued on next page*

*Figure 4 continued*

*Dclk1-GFP* basal cells in ALI culture. (**D**) Daily injection of DBZ following PR8 infection dramatically reduces tuft cell derivation in the lung parenchyma. n=4 per group. Data represent mean ± s.e.m. *p<0.05, ***p<0.001; statistical analysis by unpaired two-tailed Student's *t*-test. Scale bars, 50 µm.

The online version of this article includes the following figure supplement(s) for figure 4:

**Figure supplement 1.** Isolation and expansion of Krt5⁺ EBCs from the lungs of *Trp63^CreERT2^; R26^Ai14^* mice that were infected with PR8 virus.

Recent lineage tracing studies indicated that EBCs did not contribute meaningfully to the regenerated alveolar cells (*Kanegai et al., 2016*; *Vaughan et al., 2015*). We also injected Tmx into *KRT5-CreERT2;R26^Ai14^* mice after viral infection and did not observe contribution of lineage labeled cells to alveolar epithelium (data not shown). By contrast, we observed significant contribution of *Trp63^CreERT2^* lineage labeled cells to the alveolar epithelium when Tmx was injected after viral infection. AT2 cells were recently found to generate p63⁺ Krt5⁺ basal cells in vitro (*Kathiriya et al., 2022*). It is possible that our strategy labels these AT2 cell-derived basal cell subpopulations during regeneration. However, we did not observe *KRT5-CreERT2* lineage-labeled alveolar epithelium no matter Tmx is injected before or after viral infection. This led us to postulate that we indeed labeled a progenitor cell population that transiently expressed p63 but not Krt5 following viral infection. *Trp63^CreERT2^* lineage-labeled progenitor cells have been shown to give rise to AT1 and AT2 cells when Tmx is injected at the very early stage of mouse lung development (embryonic day 10.5) (*Yang et al., 2018*). Therefore, it is possible that subpopulations of AT1/AT2 cells regain the transcription program of fetal lung progenitors and transiently express p63 prior to becoming alveolar cells.

We observed increased contribution of *Trp63^CreERT2^*-labeled cells to the alveolar epithelium in *Trpm5^-/-^* but not *Pou2f3^-/-^* mutants, suggesting that lung regeneration is independent of the presence of ectopic tuft cells in the parenchyma. Trpm5 is a calcium-activated channel protein that induces depolarization in response to increased intracellular calcium (*Prawitt et al., 2003*). This protein is also expressed in B lymphocytes (*Sakaguchi et al., 2020*). Notably, we observed decreased accumulation of B lymphocytes in the lungs of *Trpm5^-/-^* mutants following viral infection (unpublished data, H.H. and J.Q.). It will be interesting in the future to determine whether reduced B lymphocytes facilitate lung regeneration.

In summary, we demonstrated that tuft cells are present in the airways at the early postnatal stages and later are restricted to the large airways. In response to severe injuries tuft cells expand in the airways and are ectopically present in the parenchyma where EBCs serve as their progenitor cells. Moreover, we identified *Trp63^CreERT2^* labeled alveolar epithelial cells arising during lung regeneration independent of tuft cells.

# Materials and methods

## Key resources table

| Reagent type (species) or resource | Designation | Source or reference | Identifiers | Additional information |
|---|---|---|---|---|
| Genetic reagent (*Mus musculus*) | *Trp63^CreERT2^* | *Lee et al., 2014b*; *Lee et al., 2014a* PMID:25210499 | | Dr. Jianming Xu (Baylor College of Medicine) |
| Genetic reagent (*Mus musculus*) | Tg(*KRT5-CreERT2*) | *Rock et al., 2009* PMID:19625615 | | Dr. Brigid Hogan (Duke University); A transgenic mouse strain in which human *KRT5* promoter drives CreERT2 |
| Genetic reagent (*Mus musculus*) | *Scgb1a1^CreERT^* | *Rawlins et al., 2009* PMID:19497281 | MGI:3849566 | Dr. Brigid Hogan (Duke University) |
| Genetic reagent (*Mus musculus*) | *Pou2f3^CreERT2^* | *McGinty et al., 2020* PMID:32160525 | MGI:6755141 | Dr. Jakob von Moltke (University of Washington) |
| Genetic reagent (*Mus musculus*) | Tg(*Trpm5-GFP*) | *Clapp et al., 2006* PMID:16573824 | 16573824 | Dr. Tod Clapp (Colorado State University) |
| Genetic reagent (*Mus musculus*) | *Trpm5^-/-^* | *Damak et al., 2006* PMID:16436689 | | Dr. Robert Margolskee (Mount Sinai) |

*Continued on next page*

*Continued*

| Reagent type (species) or resource | Designation | Source or reference | Identifiers | Additional information |
|---|---|---|---|---|
| Genetic reagent (*Mus musculus*) | *Pou2f3*⁻/⁻ | **Matsumoto et al., 2011** PMID:21572433 | MGI:5140071 | Dr. Keiko Abe (The University of Tokyo) |
| Genetic reagent (*Mus musculus*) | *Rbpjk*<sup>loxp/loxp</sup> | **Han et al., 2002** PMID:12039915 | MGI:3583755 | Dr. Tasuku Honjo (Kyoto University) |
| Genetic reagent (*Mus musculus*) | B6.Cg-*Gt(ROSA)26Sor*<sup>tm14(CAG-tdTomato)Hze</sup>/J | **Madisen et al., 2010** PMID:20023653 | MGI: 4436847 | Jackson Laboratories (#007914, *R26*<sup>Ai14</sup>) |
| Antibody | Anti-Dclk1 (Rabbit polyclonal) | Abcam | ab31704 | 1:200 |
| Antibody | Anti-CC10 (E-11) (Mouse monoclonal) | Santa Cruz | sc-365992 | 1:200; Scgb1a1 |
| Antibody | Anti-SCGB1A1 (Rat monoclonal) | R&D | MAB4218 | 1:500 |
| Antibody | Anti-Krt5 (Chicken polyclonal) | BioLegend | 905901 | 1:500 |
| Antibody | Anti-Krt5 (Mouse monoclonal) | Abcam | ab17130 | 1:500 |
| Antibody | Anti-Krt5 (Rabbit polyclonal) | Abcam | ab53121 | 1:500 |
| Antibody | Anti-p63 (Rabbit polyclonal) | Genetex | GTX102425 | 1:200 |
| Antibody | Anti-p63 (Mouse monoclonal) | Abcam | ab735 | 1:200 |
| Antibody | Anti-Acetylated tubulin (Mouse monoclonal) | Sigma | T7451 | 1:500 |
| Antibody | Anti-TTF1 (Rabbit monoclonal) | Abcam | ab76013 | 1:500; Nkx2.1 |
| Antibody | Anti-FOXA2 (Mouse monoclonal) | Abcam | ab60721 | 1:200 |
| Antibody | Anti-POU2F3 (Rabbit polyclonal) | Sigma | HPA019652 | 1:200 |
| Antibody | Anti-ABCA3 (Rabbit polyclonal) | Seven Hills | WRAB-70565 | 1:500 |
| Antibody | Anti-Hop (E-1) (Mouse monoclonal) | Santa Cruz | sc-398703 | 1:100; Hopx |
| Antibody | Anti- Prosurfactant Protein C (Rabbit polyclonal) | Abcam | ab90716 | 1:500; SftpC |
| Antibody | Anti-Pro-SP-C (Rabbit polyclonal) | Seven Hills | WRAB-9337 | 1:1000 |
| Antibody | Anti-tdTomato (Goat polyclonal) | Biorbyt | orb182397 | 1:1000 |
| Antibody | Anti-Pdpn (Syrian hamster monoclonal) | DSHB | 8.1.1 c | 1:500; T1α |
| Antibody | Anti- Green Fluorescent Protein (Chicken polyclonal) | Fisher (Aves Lab) | GFP1020 | 1:200; GFP |
| Antibody | Anti-Ki-67 (Rat monoclonal) | Invitrogen | 14-5698-82 | 1:50 |
| Antibody | Alexa Fluor 488 Donkey Anti-Chicken (Donkey polyclonal) | Jackson Immuno Research | 703-546-155 | 1:500 |
| Antibody | Alexa Fluor 488 Donkey Anti-Mouse (Donkey polyclonal) | Jackson Immuno Research | 715-545-151 | 1:500 |

*Continued on next page*

Continued

| Reagent type (species) or resource | Designation | Source or reference | Identifiers | Additional information |
|---|---|---|---|---|
| Antibody | Cy5 AffiniPure Donkey Anti-Rat IgG (Donkey polyclonal) | Jackson Immuno Research | 712-175-150 | 1:500 |
| Antibody | Cy3 AffiniPure Donkey Anti-Mouse IgM (Donkey polyclonal) | Jackson Immuno Research | 715-165-140 | 1:500 |
| Antibody | Alexa Fluor 488 Donkey anti-Rat IgG (Donkey polyclonal) | Invitrogen | A21208 | 1:500 |
| Antibody | Alexa Fluor 555 Donkey anti-Rabbit IgG (Donkey polyclonal) | Invitrogen | A31572 | 1:500 |
| Antibody | Alexa Fluor 555 Donkey anti-Goat IgG (Donkey polyclonal) | Invitrogen | A21432 | 1:500 |
| Antibody | Alexa Fluor 555 Donkey anti-Mouse IgG (Donkey polyclonal) | Invitrogen | A31570 | 1:500 |
| Antibody | Alexa Fluor 647 Goat anti-Chicken IgG (Goat polyclonal) | Invitrogen | A21449 | 1:500 |
| Chemical compound, drug | DBZ | Tocris | 4489 | |
| Chemical compound, drug | Wnt-C59 | Tocris | 5148 | |
| Chemical compound, drug | CHIR99021 | Tocris | 4423 | |
| Chemical compound, drug | IWR-1 | Tocris | 3532 | |
| Chemical compound, drug | Dorsomorphin | Sigma | P5499 | |
| Chemical compound, drug | Verteporfin | Sigma | SML0534 | |
| Chemical compound, drug | GDC-0449 | Selleckchem | S1082 | |
| Chemical compound, drug | Dexamethasone | Sigma | D2915 | |
| Chemical compound, drug | 3-Isobutyl-1-methylxanthine (IBMX) | Sigma | I5879 | |
| Peptide, recombinant protein | Recombinant Il-6 | Peprotech | 200–06 | |
| Peptide, recombinant protein | Recombinant Il-4 | Peprotech | 200–13 | |
| Peptide, recombinant protein | Recombinant Murine Il-13 | Peprotech | 210–13 | |
| Chemical compound, drug | 8-Bromoadenosine 3',5'-cyclic monophosphate (8-Br-cAMP) | Sigma | B5386 | |
| Chemical compound, drug | Bleomycin | Fresenius Kabi | 63323013610 | |
| Chemical compound, drug | Naphthalene | Sigma-Aldrich | 84679 | |
| Commercial assay or kit | Small Airway Epithelial Cell Medium | Lonza | CC-3118 | |
| Commercial assay or kit | Small Airway Epithelial Cell Medium | Promocell medium | C-21170 | |
| Commercial assay or kit | Complete Pneumacult-ALI medium | StemCell Technology | 05001 | |
| Other | Antigen unmasking solution, Citric acid based | Vector Laboratories | H-3300 | Antigen retrieval buffer for immunostaining |
| Other | Normal donkey serum | Jackson Immuno Research | 017-000-121 | Blocking reagent for immunostaining |

## Mouse models

All animal studies used a minimum of three mice per group. All mouse studies were approved by Columbia University Medical Center Institutional Animal Care and Use Committees (Approval protocol number AC-AABM6565). $Trp63^{CreERT2}$ (**Lee et al., 2014b**), Tg(KRT5-CreERT2) (**Rock et al., 2009**), $Rbpjk^{loxp/loxp}$ (**Han et al., 2002**), $Scgb1a1^{CreERT}$ (**Rawlins et al., 2009**), $Pou2f3^{CreERT2}$ (**McGinty et al., 2020**), Tg(Trpm5-GFP) (**Clapp et al., 2006**), $Trpm5^{-/-}$ (**Damak et al., 2006**), and $Pou2f3^{-/-}$ (**Matsumoto et al., 2011**) mouse strains were previously described. B6.Cg-$Gt(ROSA)26Sor^{tm14(CAG-tdTomato)Hze}$/J ($R26^{Ai14}$) (**Madisen et al., 2010**) mouse strain was purchased from The Jackson Laboratory (Stock #007914). All mice were maintained on a C57BL/6 and 129SvEv mixed background and housed in the

**Figure 5.** Increased generation of *Trp63^CreERT2^* labeled alveolar epithelium in *Trpm5* null mutants following PR8 infection. (**A**) *Trpm5* deletion leads to the increased presence of tdT⁺SftpC⁺Krt5⁻ cells in the lung parenchyma at 60 dpi. (**B**) Increased tdT-labeled AT1 cells in the mutant lungs as compared to controls at 60 dpi. (**C**) Whole lung airway resistance improves in *Trpm5^-/-^* mice following viral infection (left panel). Central airway resistance is also

*Figure 5 continued on next page*

*Figure 5 continued*

reduced following viral infection (right panel). n=7 for WT group, n=9 for *Trpm5*$^{-/-}$ group. Data represent mean ± s.e.m. *p<0.05, **p<0.01; statistical analysis by unpaired two-tailed Student's *t*-test. Scale bars, 100 μm (**A**), 20 μm (**B**).

The online version of this article includes the following figure supplement(s) for figure 5:

**Figure supplement 1.** Increased *Trp63*$^{CreERT2}$ labeled AT1 and AT2 cells in virus-infected *Trpm5*$^{-/-}$ mice.

mouse facility at Columbia University according to institutional guidelines. Eight–12 weeks old animals of both sexes were used in equal proportions.

## Administration of tamoxifen

Tamoxifen was dissolved in sunflower seed oil to 20 mg/mL as stock solution. For lineage analysis and genetic targeting, *Pou2f3*$^{CreERT2}$ mouse strain were administered 2 mg tamoxifen by oral gavage at days 0, 2, and 4 for control or 1 mg tamoxifen by oral gavage at days 14, 17, 20, 23, 25, 27 post-viral infection. *Trp63*$^{CreERT2}$ mice were administered with tamoxifen at a dose of 0.25 mg/g bodyweight by daily oral gavage for a total five doses as indicated. For *Scgb1a1*$^{CreERT}$ mice, a period of 10 or 21 days as indicated was used to wash out the residual tamoxifen before any further treatments.

## Injury models (influenza, bleomycin, and naphthalene)

For influenza virus infection, 260 plaque forming units (pfu) of influenza A/Puerto Rico/8/1934 H1N1 (PR8) virus were dissolved in 40 μl of RPMI medium and then pipetted onto the nostrils of anesthetized mice, whereupon mice aspirated the fluid directly into their lungs. Post procedure, mice were weighed weekly and monitored for mortality rate. For bleomycin injury, mice were anesthetized and intratracheally instilled with 4 unit/kg body weight of bleomycin hydrochloride. For naphthalene treatment, naphthalene solution (25 mg/ml) was freshly prepared before the procedure by dissolving naphthalene in sunflower seed oil. A single dose of naphthalene was delivered by intraperitoneal injection at 275 mg/kg body weight. For all procedures listed above, the administration of the same volumes of vehicle (PRMI medium or saline or sunflower seed oil) was used as control.

## Mouse EBC isolation, culture, and differentiation

*KRT5-CreERT2; R26*$^{Ai14}$ mice were infected with PR8 influenza virus and were administered with tamoxifen intraperitoneally as indicated. At 18 dpi mouse peripheral lungs were dissected and dissociated according to the protocol as previously described (*Barkauskas et al., 2013*; *Rock et al., 2011*). tdT$^+$ cells were sorted by FACS and cultured using the protocol as previously reported (*Mou et al., 2016*), and the protocol for inducing the differentiation of basal-like cells was previously described (*Feldman et al., 2019*; *Mou et al., 2016*). Air-liquid interface (ALI) culture was used to test the effects of the major signaling pathway inhibitors on the differentiation of EBCs towards tuft and mucous cell lineages. Moreover, tracheal basal cells isolated from *Trpm5-GFP* mice were cultured and tested for drug effects in ALI. The tested inhibitors include the BMP signaling inhibitor Dorsomorphin (5 μM), YAP signaling inhibitor verteporfin (100 nM), NOTCH signaling inhibitor DBZ (2 μM), SHH signaling inhibitor GDC-0449 (1 μM), WNT signaling inhibitor IWR-1 (5 μM), WNT signaling activator CHIR99021 (2 μM) which inhibits glycogen synthase kinase (GSK) 3, IL-6 (10 ng/ml), IL-4 (10 ng/ml) and IL-13 (10 ng/ml) were also used to treat ALI culture of EBCs.

## Treatment of mice with the Porcupine inhibitor Wnt-C59 and the γ secretase inhibitor DBZ

Wnt-C59 was resuspended by sonication for 20 minutes in a mixture of 0.5% methylcellulose and 0.1% tween-80. Wnt-C59 (10 mg/kg body weight) or vehicle was administrated via oral gavage from day 14 to 29 post-viral infection (n=4 for each group). For DBZ administration, either vehicle or DBZ was administered intranasally at 30 mmol/kg body weight (n=4 per group) from day 14 to 29 post-viral infection. DBZ was suspended in sterile PBS mixed with 2.5 μg/g body weight dexamethasone.

## Tissue and ALI culture processing and immunostaining

Human and mouse lung tissues were fixed in 4% paraformaldehyde (PFA) at 4°C overnight and then dehydrated and embedded in paraffin for sections (5–7 μm). ALI culture membranes were fixed with

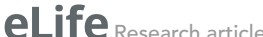

**Figure 6.** A complete loss of tuft cell does not increase the generation of *Trp63*<sup>CreERT2</sup> labeled alveolar cells following PR8 infection. (**A**) Extensive Krt5⁺SftpC⁻ EBCs are present in the parenchyma of control (*Trp63*<sup>CreERT2</sup>;*Pou2f3*<sup>+/-</sup>;*R26*<sup>Ai14</sup>) and mutant (*Trp63*<sup>CreERT2</sup>;*Pou2f3*<sup>-/-</sup>;*R26*<sup>Ai14</sup>) lungs when examined at 60 dpi following PR8 infection. (**B**) Representative areas show *Trp63*<sup>CreERT2</sup> labeled AT2 cells in both control and mutant lungs. Ns: not significant; statistical analysis by unpaired two-tailed Student's *t*-test. Scale bar, 50 µm (**A**), 100 µm (**B**).

4% PFA for direct wholemount staining or were embedded in OCT for frozen sections. All slides and membranes were stained following the protocol reported previously (*Feldman et al., 2019*; *Mou et al., 2016*). The primary antibodies used for immunostaining are listed in the Resources Table.

## Pulmonary function assessment

*Trpm5*[+/-] and *Trpm5*[-/-] mice ( +/- days after viral injury) were anesthetized with pentobarbital (50 mg/kg, i.p.). After surgical anesthesia was achieved, a tracheotomy was performed for the insertion of an 18 G cannula, and mice were immediately connected to a flexiVent (SciReq, Montreal, QC, Canada) with an FX1 module and an in-line nebulizer. Body temperature was monitored and maintained with a thermo-coupled warming blanket and rectal temperature probe. Heart rate was monitored by EKG (electrocardiography). Mice were mechanically ventilated with a tidal volume of 10 mg/kg, frequency of 150 breaths/min, and a positive end expiratory pressure of 3 mmHg. Muscle paralysis was achieved with succinylcholine (10 mg/kg, i.p.) to prevent respiratory effort and to eliminate any contribution of chest wall muscle tone to respiratory measurements. By using the forced oscillation technique, baseline measurements of lung resistance (Rrs and Rn, representing total and central airway resistance) were performed. Resistance was then measured during nebulization of increasing concentrations of methacholine (10 s nebulization, 50% duty cycle). Methacholine dissolved in PBS was nebulized at 0, 6.25, 12.5, 25, and 50 mg/ml and resistance (Rrs and Rn) was measured after each concentration. Values for all measurements represent an average of triplicate measurements. Statistical significance was established by comparing the area under the curve for each mouse.

## COVID-19 lung specimens

The lung specimens from deceased COVID-19 patients with short post-mortem interval (PMI) (2.5–9 hr) were obtained from the Biobank at Columbia University Irving Medical Center. All experiments involving human samples were performed in accordance with the protocols approved by the Institutional Review Boards at Columbia University Irving Medical Center.

## Microscopic imaging and quantification

Slides were visualized using a ZeissLSM T-PMT confocal laser-scanning microscope (Carl Zeiss). The staining of cells on culture dishes and the staining on transwell membranes were visualized with the Olympus IX81 inverted fluorescence microscope. For quantification of lineage tracing, the lung sections were tiled scanned with 20 X images from at least three mice for each genotype. Cells were counted from at least five sections per mouse including at least three individual lung lobes. The production of various airway epithelial cell types was counted and quantified on at least 5 random fields of view with a 10 X or a 20 X objective, and the average and standard deviation was calculated.

## Quantification and statistical analysis

Data are presented as means with standard deviations of measurements unless stated otherwise (n≥3). Statistical differences between samples are assessed with Student two-tailed T-test. p-Values below 0.05 are considered significant (*p<0.05, **p<0.01, ***p<0.001).

## Acknowledgements

We thank the colleagues in the Que laboratory for critical input of the study. We also thank Columbia University COVID-19 Hub and the autopsy pathologists who have been fighting on the frontlines and provided us short post-mortem interval (PMI) specimens. Funding: This work is partly supported by R01HL152293, R01HL159675, R01DK120650, R01DK100342 (to JQ), Cystic Fibrosis Foundation Research Grant MOU19G0 (to HM), Harvard Stem Cell Institute Seed Grant (SG-0120-19-00), Charles H Hood Foundation Child Health Research Award (to HM), Discovery Award from the Department of Defense (W81XWH-21-1-0196 to HH) and R21AI163753 (to HH).Research reported in this publication was supported in part by the Office of the Director, NIH Award 1S10OD032447, S10OD020056, P30CA013696 and P30DK132710.

# Additional information

## Funding

| Funder | Grant reference number | Author |
|---|---|---|
| National Heart, Lung, and Blood Institute | R01HL152293 | Jianwen Que |
| National Heart, Lung, and Blood Institute | R01HL159675 | Jianwen Que |
| National Institute of Diabetes and Digestive and Kidney Diseases | R01DK120650 | Jianwen Que |
| National Institute of Diabetes and Digestive and Kidney Diseases | R01DK100342 | Jianwen Que |
| Cystic Fibrosis Foundation | MOU19G0 | Hongmei Mou |
| Harvard Stem Cell Institute | SG-0120-19-00 | Hongmei Mou |
| Charles H. Hood Foundation | | Hongmei Mou |
| U.S. Department of Defense | W81XWH-21-1-0196 | Huachao Huang |
| National Institute of Allergy and Infectious Diseases | R21AI163753 | Huachao Huang |

The funders had no role in study design, data collection and interpretation, or the decision to submit the work for publication.

## Author contributions

Huachao Huang, Conceptualization, Formal analysis, Funding acquisition, Methodology, Writing - original draft, Writing - review and editing; Yinshan Fang, Formal analysis, Methodology, Writing - review and editing; Ming Jiang, Yihan Zhang, Jana Biermann, Johannes C Melms, Jennifer A Danielsson, Ying Yang, Jia Liu, Formal analysis, Methodology; Li Qiang, Yiwu Zhou, Manli Wang, Zhihong Hu, Timothy C Wang, Anjali Saqi, Jie Sun, Ichiro Matsumoto, Wellington V Cardoso, Charles W Emala, Jian Zhu, Benjamin Izar, Writing - review and editing; Hongmei Mou, Formal analysis, Supervision, Funding acquisition, Investigation, Methodology, Writing - review and editing; Jianwen Que, Conceptualization, Supervision, Funding acquisition, Investigation, Writing - review and editing

## Author ORCIDs

Yinshan Fang ⬤ http://orcid.org/0000-0001-9249-4498
Jana Biermann ⬤ http://orcid.org/0000-0002-8907-4633
Johannes C Melms ⬤ http://orcid.org/0000-0002-5410-6586
Ying Yang ⬤ http://orcid.org/0000-0002-4197-6216
Li Qiang ⬤ http://orcid.org/0000-0001-8322-1797
Zhihong Hu ⬤ http://orcid.org/0000-0002-1560-0928
Wellington V Cardoso ⬤ http://orcid.org/0000-0002-8868-9716
Benjamin Izar ⬤ http://orcid.org/0000-0003-2379-6702
Jianwen Que ⬤ http://orcid.org/0000-0002-6540-6701

## Ethics

All animal studies used a minimum of three mice per group. Mouse studies were approved by Columbia University Medical Center Institutional Animal Care and Use Committees (Approval protocol number AC-AABM6565).

## Decision letter and Author response

Decision letter https://doi.org/10.7554/eLife.78217.sa1
Author response https://doi.org/10.7554/eLife.78217.sa2

## Additional files

### Supplementary files

• Transparent reporting form

### Data availability

Data Availability: All data are available in the main text or the supplementary materials and deposited to Dryad https://doi.org/10.5061/dryad.0vt4b8h1w.

The following dataset was generated:

| Author(s) | Year | Dataset title | Dataset URL | Database and Identifier |
|---|---|---|---|---|
| Que J | 2022 | Alveolar regeneration following viral infection is independent of tuft cells | https://dx.doi.org/10.5061/dryad.0vt4b8h1w | Dryad Digital Repository, 10.5061/dryad.0vt4b8h1w |

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
