## [Editor Report]

In this manuscript, the authors describe the ectopic tuft cells that were derived from lineage-tagged Krt5+ Trp63+ cells post influenza virus infection. These tuft cells do not appear to proliferate or give rise to other lineages in the lung parenchyma. They then claim that Wnt inhibitors increase the number of tuft cells while inhibiting Notch signalling decreases the number of tuft cells within Krt5+ pods after infection in vitro and in vivo. The authors further show that genetic deletion of Trpm5 results in an increase in AT2 and AT1 cells in p63CreER lineage-tagged cells compared to control. Lastly, they demonstrate that depletion of tuft cells caused by genetic deletion of Pou2f3 has no effect on the derivation of AT1 and AT2 cells from the p63+ Krt5- progenitor population, implying that tuft cells play no functional role in this process. This study provides new insights into the role of tuft cells and p63+Krt5- progenitor cells in lung repair.

---

## [Decision Letter]

**Decision letter after peer review:**

Thank you for submitting your article "Alveolar regeneration following viral infection is independent of tuft cells" for consideration by *eLife*. Your article has been reviewed by 3 peer reviewers, and the evaluation has been overseen by a Reviewing Editor and Paul Noble as the Senior Editor. The reviewers have opted to remain anonymous.

The reviewers have discussed their reviews with one another, and the Reviewing Editor has drafted this to help you prepare a revised submission. All reviewers agree on the importance of the primary observation that Tuft cells do not contribute to alveolar regeneration after injury. However, substantial concerns regarding the preliminary nature of some of the findings were highlighted. Please address the concerns raised below.

Essential revisions:

1) Please analyze which cells p63-CreERT2 labels immediately after PR8 and tamoxifen treatment. Are all the tdTomato labeled cells also Krt5 and p63 positive or are some alveolar epithelial cells or other airway cell types also labeled?

2) Please also show if p63-CreERT2 labels any cells in the adult lung parenchyma in the absence of injury after tamoxifen treatment.

3) Please analyze if p63-CreERT2 labels any cells with tdTomato in the absence of injury or after PR8 infection but without tamoxifen treatment.

4) Please analyze when after PR8 infection do the first p63-CreERT2 labeled tdTomato positive alveolar epithelial cells appear.

5) A clonal analysis of p63-CreERT2 labeled cells using a confetti reporter might also help interpret the origin of p63-CreERT2 labeled cells.

6) Lastly could the authors compare the single-cell RNAseq transcription profile of p63-CREERT2 labeled cells immediately after PR8 and tamoxifen treatment and also at 60dpi. A pseudotime analysis and trajectory interference analysis could help elucidate the identity of p63-CreERT2 labeled cells that are actually not ectopic basal progenitor cells.

7) Origin of tuft cells: Although the authors showed the emergence of ectopic tuft cells derived from labelled p63+ cells after infection, it cannot be ruled out that pre-existing p63+Krt5- intrapulmonary progenitors, as previously reported, can also contribute to tuft cell expansion (Rane et al. 2019; by labelling p63+ cells prior to infection, they showed that the majority of ectopic tuft cells are derived from p63+ cells after viral infection). It would be more informative if the authors show the differentiation of tuft cells derived from p63+Krt5+ cells by tracing Krt5+ cells after infection, which will tell us whether ectopic tuft cells are differentiated from ectopic basal cells within Krt5+ pods induced by virus infection.

8) Mechanisms of tuft cell differentiation: The authors tried to determine which signalling pathways regulate the differentiation of tuft cells from p63+ cells following infection. Although Wnt/Notch inhibitors affected the number of tuft cells derived from p63+ labelled cells, it remains unclear whether these signals directly modulate differentiation fate. The authors claimed that Wnt inhibition promotes tuft cell differentiation from ectopic basal cells. However, in Figure 3B, Wnt inhibition appears to trigger the expansion of p63+Krt5+ pod cells, resulting in increased tuft cell differentiation rather than directly enhancing tuft cell differentiation. Further, in Figure 3D, Notch inhibition appears to reduce p63+Krt5+ pod cells, resulting in decreased tuft cell differentiation. Importantly, a previous study has reported that Notch signalling is critical for Krt5+ pod expansion following influenza infection (Vaughan et al. 2015; Xi et al. 2017). Notch inhibition reduced Krt5+ pod expansion and induced their differentiation into Sftpc+ AT2 cells. One of the key findings in this manuscript is that Wnt and Notch signaling play a role in Tuft cell specification. All current experiments are based on pharmacological modulation. These need to be substantiated using genetic gain loss of function models. In order to address the direct effect of Wnt/Notch signalling in the differentiation process of tuft cells from EBCs, the authors should provide a more detailed characterization of cellular composition (Krt5+ basal cells, club cells, ciliated cells, AT2 and AT1 cells, etc.) and activity (proliferation) within the pods with/without inhibitors/activators.

9) Impact of Trpm5 deletion in p63+ cells: It is interesting that Trpm5 deletion promotes the expansion of AT2 and AT1 cells derived from labelled p63+ cells following infection. It would be informative to check whether Trpm5 regulates Hif1a and/or Notch activity which has been reported to induce AT2 differentiation from ectopic basal cells (Xi et al. 2017). Although the authors stated that there was no discernible reduction in the size of Krt5+ pods in mutant mice, it would be interesting to investigate the relationship between AT2/AT1 cell retaining pods and the severity of injury (e.g. large Krt5+ pods retain more/less AT2/AT1 cells compared to small pods. What about other cell types, such as club and goblet cells, in Trpm5 mutant pods? Again, it cannot be ruled out that pre-existing p63+Krt5- intrapulmonary progenitor cells can directly convert into AT2/AT1 cells upon Trpm5 deletion rather than p63+Krt5+ cells induced by infection.

10) Quantification information and method: Overall, the quantification method should be clarified throughout the manuscript. Further, in the method section, the authors stated that the production of various airway epithelial cell types was counted and quantified on at least 5 "random" fields of view. However, virus infection causes spatially heterogeneous injury, resulting in a difficult to measure "blind test". The authors should address how they dealt with this issue.

---

## [Author Response]

Essential revisions:1) Please analyze which cells p63-CreERT2 labels immediately after PR8 and tamoxifen treatment. Are all the tdTomato labeled cells also Krt5 and p63 positive or are some alveolar epithelial cells or other airway cell types also labeled?

We thank the reviewer for the question. To answer the reviewer’s question, we performed PR8 infection (250 pfu) on three *Trp63-CreERT2*;*R26tdT* mice and TMX treatment at days 5 and 7 post viral infection. We didn't perform TMX injection immediately as the mice were sick at a few days post infection. The lung samples were collected at 14 dpi. We observed that tdT^+^ cells are present in the airways (Author response image 1, B), and it appears that the lineage labeled cells (tdT^+^) include club cells (CC10^+^) that are underlined by tdT+Krt5+ basal cells (Author response image 1). We think that these labeled basal cells give rise to club cells. However, we also noticed that rare club cells and ciliated cells (FoxJ1^+^) are labeled by tdT in the areas absent of surrounding tdT^+^ basal cells (Author response image 1). Moreover, a minor population of tdT^+^ SPC^+^ cells are present in the terminal airways that were disrupted by viral infection (Author response image 1 and D). We did not see any pods formed in this experiment and we did not observe any tdT^+^ cells in the intact alveoli (uninjured area).

**Author response image 1. sa2fig1:** *Trp63-CreERT2* lineage labeled cells in the airways but not alveoli when Tamoxifen was induced at day 5 and 7 after PR8 H1N1 viral infection. *Trp63-CreERT2*;*R26-tdT* mice were infected with PR8 at 250 pfu and Tmx were delivered at a dose of 0.25 mg/g bodyweight by oral gavage. Lung samples were collected and analyzed at 14 dpi. Stained antibodies are as indicated. Scale bar: 100 µm.

2) Please also show if p63-CreERT2 labels any cells in the adult lung parenchyma in the absence of injury after tamoxifen treatment.

Dr. Wellington Cardoso’s group demonstrated that *Trp63-CreERT2* only labels very few cells in the airways but not the lung parenchyma in the absence of injury after tamoxifen treatment (Yang et al., 2018). Dr. Ying Yang has revisited the data and she did not observe any labeling in the lung parenchyma (n = 2).

3) Please analyze if p63-CreERT2 labels any cells with tdTomato in the absence of injury or after PR8 infection but without tamoxifen treatment.

We performed the experiment and didn't observe any labeled cells in the lung parenchyma without Tamoxifen treatment (n = 4).

4) Please analyze when after PR8 infection do the first p63-CreERT2 labeled tdTomato positive alveolar epithelial cells appear.

We administered tamoxifen at day 5 and 7 after PR8 infection and harvested lung tissues at day 14. As shown in Figure 1, we observed a few tdT^+^ SPC^+^ cells in the terminal airways that are disrupted by viral infection. Notably, we did not observe any lineage labeled cells in the intact alveoli (uninjured) in this experiment.

5) A clonal analysis of p63-CreERT2 labeled cells using a confetti reporter might also help interpret the origin of p63-CreERT2 labeled cells.

We thank the reviewer for the suggestion. Our new data demonstrate that a rare population of SPC^+^tdT^+^ cells are present in the disrupted terminal airways of *Trp63-CreERT2;R26tdT* mice. Our data in the original manuscript and the new data suggest that the initial SPC^+^;tdT^+^ cells are rare because we have to administrate multiple doses of Tamoxifen to label them. Given the less labeling efficiency of confetti than R26tdT mice, it is possible we will not be able to label these SPC^+^ cells. Moreover, our original manuscript clearly shows individual clones of SPC^+^tdT^+^ cells in the regenerated lung, and they do not seem to compose of multiple clones. Therefore we think that use of confetti mice may not add new information.

6) Lastly could the authors compare the single-cell RNAseq transcription profile of p63-CREERT2 labeled cells immediately after PR8 and tamoxifen treatment and also at 60dpi. A pseudotime analysis and trajectory interference analysis could help elucidate the identity of p63-CreERT2 labeled cells that are actually not ectopic basal progenitor cells.

We appreciated the reviewer’s suggestion and agree that single cell RNA sequencing with pseudotime analysis can provide further information regarding the origin of the lineage labeled alveolar cells of *Trp63-CreERT2;R26tdT* mice. That said, our new data clearly show that *KRT5-CreER* lineage labeled cells do not give rise to AT_1/2_ cells as previously described (Kanegai et al., 2016; Vaughan et al., 2015), suggesting that the ectopic basal progenitor cells do not generate alveolar cells. By contrast, *Trp63-CreERT2* lineage labeled cells do give rise to AECs, suggesting that this p63^+^ cell population capable of generating AECs are different from Krt5^+^ ectopic basal progenitor cells. Our single cell core has an extremely long waiting list due to the pandemic and we hope that our new findings are enough to address the reviewer’s concern without the need of single cell analysis.

7) Origin of tuft cells: Although the authors showed the emergence of ectopic tuft cells derived from labelled p63+ cells after infection, it cannot be ruled out that pre-existing p63+Krt5- intrapulmonary progenitors, as previously reported, can also contribute to tuft cell expansion (Rane et al. 2019; by labelling p63+ cells prior to infection, they showed that the majority of ectopic tuft cells are derived from p63+ cells after viral infection). It would be more informative if the authors show the differentiation of tuft cells derived from p63+Krt5+ cells by tracing Krt5+ cells after infection, which will tell us whether ectopic tuft cells are differentiated from ectopic basal cells within Krt5+ pods induced by virus infection.

We followed the reviewer’s suggestion and performed lineage tracing with *Trp63-CreERT2*;*R26-tdT* mice. Mice were infected with PR8 virus at day0, and Tmx was injected daily between day 14 and 18, and the lung samples were collected at day 30. Our lineage tracing data clearly demonstrated that the majority of tuft cells (Dclk1^+^) are lineage labeled (Figure 2E). Considering the label efficiency by Tmx, we think that the major source of tuft cells are p63^+^Krt5^+^ pod cells (EBCs).

8) Mechanisms of tuft cell differentiation: The authors tried to determine which signalling pathways regulate the differentiation of tuft cells from p63+ cells following infection. Although Wnt/Notch inhibitors affected the number of tuft cells derived from p63+ labelled cells, it remains unclear whether these signals directly modulate differentiation fate. The authors claimed that Wnt inhibition promotes tuft cell differentiation from ectopic basal cells. However, in Figure 3B, Wnt inhibition appears to trigger the expansion of p63+Krt5+ pod cells, resulting in increased tuft cell differentiation rather than directly enhancing tuft cell differentiation. Further, in Figure 3D, Notch inhibition appears to reduce p63+Krt5+ pod cells, resulting in decreased tuft cell differentiation. Importantly, a previous study has reported that Notch signalling is critical for Krt5+ pod expansion following influenza infection (Vaughan et al. 2015; Xi et al. 2017). Notch inhibition reduced Krt5+ pod expansion and induced their differentiation into Sftpc+ AT2 cells. One of the key findings in this manuscript is that Wnt and Notch signaling play a role in Tuft cell specification. All current experiments are based on pharmacological modulation. These need to be substantiated using genetic gain loss of function models. In order to address the direct effect of Wnt/Notch signalling in the differentiation process of tuft cells from EBCs, the authors should provide a more detailed characterization of cellular composition (Krt5+ basal cells, club cells, ciliated cells, AT2 and AT1 cells, etc.) and activity (proliferation) within the pods with/without inhibitors/activators.

We thank the reviewer’s constructive suggestions, and we apologize for not clearly explaining our quantitative findings. The quantification data in Figure 3B (revised Figure 4B) are presented as the number of tuft cells per mm^2^ Krt5^+^ pod area.

To further confirm the role of Notch/Wnt inhibition in tuft cell derivation, we performed genetic studies, generating *KRT5-CreER*;*Rbpjk^f/f^*;*R26-tdT* mice. As shown in the following figure, immunostaining on multiple cell markers showed that *Rbpjk* knockout consistently blocks tuft cell differentiation from pod cells (Figure 4—figure supplement 1). Due to resource limitation, we couldn’t perform *β* -catenin knockout experiments at this moment.

9) Impact of Trpm5 deletion in p63+ cells: It is interesting that Trpm5 deletion promotes the expansion of AT2 and AT1 cells derived from labelled p63+ cells following infection. It would be informative to check whether Trpm5 regulates Hif1a and/or Notch activity which has been reported to induce AT2 differentiation from ectopic basal cells (Xi et al. 2017). Although the authors stated that there was no discernible reduction in the size of Krt5+ pods in mutant mice, it would be interesting to investigate the relationship between AT2/AT1 cell retaining pods and the severity of injury (e.g. large Krt5+ pods retain more/less AT2/AT1 cells compared to small pods. What about other cell types, such as club and goblet cells, in Trpm5 mutant pods? Again, it cannot be ruled out that pre-existing p63+Krt5- intrapulmonary progenitor cells can directly convert into AT2/AT1 cells upon Trpm5 deletion rather than p63+Krt5+ cells induced by infection.

We thank the reviewer for the comments and suggestions. Our new data using *KRT5-CreER* mouse line confirmed that pod cells (Krt5^+^) do not contribute to AT2/AT1 cells, consistent with previous studies (Kanegai et al., 2016; Vaughan et al., 2015). Our data also show that *p63-CreER* lineage labeled AT2/AT1 cells are separated from pod cell area, suggesting pod cells and these AT2/AT1 cells are generated from different cell of origin. We also checked the Notch activity in pod cells in *Trpm5^-/-^* mice, and some pod cell-derived cells are Hes1 positive, whereas some are Hes1 negative (Author response image 2). As indicated in *discussion* we think that AT2/AT1 cells are possibly derived from pre-existing AT2 cells that transiently express p63 after PR8 infection. It will be interesting to test whether Trpm5 regulates Hif1a in this population (p63^+^,Krt5^-^), and this will be our next plan.

**Author response image 2. sa2fig2:** Representative area staining in *Trpm5^-/-^* mice at 30 dpi. Area 1: Notch signaling is active (Hes1^+^, arrows) in pod cells following viral infection. Area 2: pod cells exhibit reduced Notch activities. Note few Hes1^+^ cells in pods (arrows). Scale bar: 50 µm.

10) Quantification information and method: Overall, the quantification method should be clarified throughout the manuscript. Further, in the method section, the authors stated that the production of various airway epithelial cell types was counted and quantified on at least 5 "random" fields of view. However, virus infection causes spatially heterogeneous injury, resulting in a difficult to measure "blind test". The authors should address how they dealt with this issue.

We clarified that quantification method as suggested. For the in vitro cell culture assays on the signaling pathways, we took pictures from at least five random fields of view for quantification. For lung sections, we tile-scanned the lung sections including at least three lung lobes and performed quantification.